# Diallyl Trisulfide Promotes Placental Angiogenesis by Regulating Lipid Metabolism and Alleviating Inflammatory Responses in Obese Pregnant Mice

**DOI:** 10.3390/nu14112230

**Published:** 2022-05-26

**Authors:** Miaomiao Wang, Zhaoyu Wang, Yueyue Miao, Hongkui Wei, Jian Peng, Yuanfei Zhou

**Affiliations:** 1Department of Animal Nutrition and Feed Science, College of Animal Science and Technology, Huazhong Agricultural University, Wuhan 430070, China; 15212192412@163.com (M.W.); 15301046182@163.com (Z.W.); 18215120167@163.com (Y.M.); weihongkui@mail.hzau.edu.cn (H.W.); 2The Cooperative Innovation Center for Sustainable Pig Production, Wuhan 430070, China

**Keywords:** maternal obesity, hydrogen sulfide, placental angiogenesis, inflammation response, lipid metabolism

## Abstract

The placental tissue serves as an exchanger between the mother and the fetus during pregnancy in mammals. Proper placental angiogenesis is central to the health of both the mother and the growth and development of the fetus. Maternal obesity is associated with impaired placental function, resulting in restricted placental blood vessel development and fetal developmental disorders. Hydrogen sulfide (H_2_S) is a ubiquitous second messenger in cells that has many biological effects such as promoting angiogenesis, anti-inflammation, anti-oxidation and promoting lipid metabolism. However, in the case of maternal obesity, whether H_2_S can be used as an important signaling molecule to regulate body metabolism, alleviate placental inflammation levels and promote placental angiogenesis is still unclear. In this study, diallyl trisulfide (DATS), which is a well-known H_2_S donor, was derived from garlic and used to treat obese pregnant mice induced by a high-fat diet, to determine its effects on lipid metabolism and inflammation, as well as placental morphology and placental angiogenesis. Here, we show that DATS treatment increased litter size and alive litter size. DATS improved the H_2_S level in the serum and placenta of the mice. In addition, DATS treatment improved insulin resistance and lipid metabolism, reduced the inflammatory response and alleviated placental vascular dysplasia caused by obesity in obese mice. In summary, our research revealed that H_2_S is an important signaling molecule in vivo, which can regulate placental angiogenesis and improve the reproductive performance in maternal obesity. The addition of H_2_S donor DATS during pregnancy promoted placental angiogenesis by regulating lipid metabolism and alleviating inflammatory responses in obese pregnant mice.

## 1. Introduction

Obesity is a major risk factor for various metabolic syndromes and complications, including cardiovascular disease, type 2 diabetes and some types of cancer [1,2,3]. Maternal obesity during pregnancy may increase the incidence of adverse maternal and fetal outcomes, including preeclampsia, premature delivery, stillbirth, congenital malformations and macrosomia [4]. As an important place for the exchange of nutrients and oxygen between the mother and the fetus, the placenta provides all the nutrients needed for the growth of the developing fetus [5,6]. Studies show that maternal obesity can lead to impaired placental function, thus affecting placental angiogenesis [7]. Moreover, abnormal placental vascular development is associated with placental vascular diseases, affecting fetal intrauterine development, and seriously endangers the safety of mother and fetus [8]. Our previous studies found that sows’ obesity during pregnancy significantly inhibited the expression of vascular endothelial growth factor (VEGF), and decreased the placental vascular density, resulting in reduced placental vascular, resulting in reduced placental vascular development, poor placental function and compromised fetal growth and development, thus increasing the generation of weak offspring with body weight ≤0.9 kg [9]. Therefore, the maintenance of normal angiogenesis and placental function in maternal obesity is the key factor to ensure the growth and development of offspring [10].

As a ubiquitous second messenger in cells, H_2_S can freely cross the cell membrane in a variety of systems and plays a variety of biological functions as an endogenous signaling molecule in vivo, including the promotion of angiogenesis, anti-inflammation, anti-oxidation, the inhibition of cell apoptosis and the regulation of ion channels, etc. [11,12,13]. In animal studies, H_2_S has been found to promote wound healing in ischemic diabetic lower limb, and to increase the expression of proangiogenic factors and angiogenesis in the ischemic adductor muscle [14]. In vitro studies have also shown that H_2_S produced by trophoblasts can promote the angiogenesis of placental artery endothelial cells [15]. Moreover, in terms of anti-inflammatory, studies indicate that H_2_S can reduce cigarette smoking-induced inflammation by inhibiting the phosphorylation of ERK1/2, JNK and p38 MAPKs and negatively regulating the activation of NF-κB, so as to protect smoking rats from pulmonary fibrosis [16]. In terms of lipid metabolism, H_2_S can reduce weight gain and fat accumulation by inhibiting fat production and promoting lipolysis [17]. Lipid metabolism, inflammation, and angiogenesis are not independent of each other. Studies have shown that increased lipid accumulation in the placenta leads to increased macrophage infiltration and decreased placental blood flow, leading to hypoxia and exacerbating placental inflammation, and altering nutrient transport, while chronic inflammation caused by obesity can lead to an imbalance of placental angiogenesis [18,19,20,21,22]. Therefore, H_2_S can also stimulate angiogenesis by regulating the lipid metabolism of obese individuals and activating the phosphatidylinositol 3-kinase (PI3K) and Akt pathways [19,23].

In this study, DATS was selected as the hydrogen sulfide donor. Compared with traditional NaHS and GYY4137, DATS can maintain a high concentration of H_2_S for a longer time and has better anti-inflammatory and antioxidant effects [24]. We explored the potential effects of H_2_S donor DATS on obese mice induced by a high-fat diet. We found that DATS treatment improved maternal metabolic health and promoted fetal growth and development by enhancing placental vascular development.

## 2. Materials and Methods

### 2.1. Animals and Diets

In this study, one-hundred female C57Bl/6J mice (6–8 weeks of age) were purchased from Hubei Experimental Animal Research Center, and animal procedures were performed according to the Guidelines of the Institutional Animal Care and Use Committees at the Huazhong Agricultural University; the ethical approval number is HZAUMO-2016-054. The animals were housed in stainless steel cages at room temperature (24 ± 2 °C), with a 12 h light/dark cycle. They were fed a regular chow for a week to acclimatize them to the animal facilities, weighed, and randomly divided into two groups. One group was fed regular chow (control group, NFD, *n* = 40) and the other group received a 60% kcal high-fat diet (HFD, *n* = 60) (TROPHIC Animal Feed High-Tech Co. Ltd., Nantong, China, TP 23300) for 13 weeks to induce obesity.

### 2.2. Animal Experiment

After 13 weeks, the mice in the HFD group had gained more than 25% of their initial body weight, showing HFD-induced obesity [25,26]. Then, female mice were mated with 10-week-old male mice overnight. Pregnancy was determined by the presence of a vaginal plug and was assigned the title E0.5. Next, the NFD group and HFD group were divided into the following two subgroups: the control group (NFD; *n* = 9, fed regular chow, with PBS administered by gavage), the treatment group (NFD + DATS; *n* = 11, with 50 mg/kg/d DATS administered by gavage), the HFD group (*n* = 11, fed a high-fat diet, with PBS administered by gavage), and the high-fat treatment group (HFD + DATS; *n* = 10, with 50 mg/kg/d DATS (B25320, Source Leaf Biological, Shanghai, China) administered by gavage) [27]. The female mice were treated by gavage from E0.5 to E18.5. During pregnancy, the body weight and feed intake of the female mice were recorded at 9:00 a.m. every day.

### 2.3. Sample Collection

At E18.5, the mice were weighed at 8:30 a.m. and fasted for 6 hours. After ether inhalation and anesthesia, blood samples from the orbit were taken and the neck was dislocated. After standing at room temperature for 30 min, the blood samples were centrifuged at 4000 r/min for 10 min at 4 °C. The serum was separated and stored in a refrigerator at −80 °C. The placenta and fetus were taken by cesarean section; each fetus and its corresponding placenta were separated and weighed. Tissues (including inguinal white adipose tissue (IWAT), epididymal white adipose tissue (EWAT) and brown adipose tissue (BAT)) were collected, weighed, snap-frozen and stored at –80 °C. Placental tissue samples were snap-frozen in liquid nitrogen storage at −80 °C or fixed in 4% paraformaldehyde for histology and immunohistochemical analysis.

### 2.4. Intraperitoneal Glucose Tolerance Test (IPGTT)

When the mice were fattened successfully, 12 mice were randomly selected from each group for intraperitoneal glucose tolerance tests (IPGTTs). The same as for the above groups, the HFD group and the NFD group were divided into HFD, HFD + DATS, NFD and NFD + DATS subgroups, according to whether DATS was given by gavage, and six mice in each group were given the same dose. The mice were fasted for 6 h at 8:30 a.m. before injection with 2 g/kg glucose at D6 [28].

### 2.5. Analysis of Lipid Levels in Maternal Serum and Placenta

Maternal serum triglyceride (TG), low-density lipoprotein cholesterol (LDL-C), high-density lipoprotein cholesterol (HDL-C) and total cholesterol (T-CHO) were detected using an Automatic biochemical analyzer (BS-240, Shenzhen Mindray Bio-medical Electronics Co. LTD, Shenzhen, China). Placental tissues were homogenized on ice with 100 mM KH_2_PO_4_ buffer (pH 7.4) (tissue weight: buffer = 1 g: 10 mL), and the supernatant was taken by centrifuging at 3000 r/min for 10 min at 4 °C, and then stored at −80 °C for later use. Placental homogenate triglyceride (TG), low-density lipoprotein cholesterol (LDL-C), high-density lipoprotein cholesterol (HDL-C) and total cholesterol (T-CHO) in the maternal serum and placenta were detected using kits (Nanjing Jiancheng Bioengineering Institute, Nanjing, China).

### 2.6. Analysis of Hydrogen Sulfide in Maternal Serum and Placenta

Firstly, 75–100 μL of maternal serum or 100 μL of placental homogenate was placed in a 1.5 mL EP tube, 100 mM KH_2_PO_4_ buffer (pH 7.4) was added to fill up the tube to 500 μL, followed by 250 μL of zinc acetate with a mass/volume (*w/v*) ratio of 1%, and it was incubated at 37 °C for 2.5 h. Secondly, 250 μL of 10% (*w/v*) trichloroacetic acid was added to the reaction mixture, and after shaking violently and centrifuging at 12,000 r/min for 10 min, the supernatant was taken. The supernatant was then mixed with 100 μL of FeCl_3_ (30 mmol/L in 1.2 mol/L HCl) and 100 μL of N, N-dimethyl-p-phenylenediamine sulfate (20 mmol/L in 7.2 mol/L HCl) in a test tube, preheated at room temperature 0.5 h in advance. The mixture was incubated at room temperature for 15 min and 200 μL was taken for use in a microplate analyzer at 670 nm to detect the absorbance value. Finally, we made the standard curve using NaHS (0–200 μmol/L) [29,30].

### 2.7. RNA Extraction and Quantitative Real-Time PCR

Total RNA from E18.5 placental tissues was extracted using TRizol reagent (Invitrogen, Waltham, MA, USA). Then, 2 μg total RNA was converted to cDNA using the Revertra ACE qPCR RT Kit (Toyobo, Osaka, Japan), according to the manufacturer’s protocol. Quantitative real-time PCR was performed using iQ SYBR green Supermix (BioRad, Hercules, CA, USA) on a CFX 384 Touch qPCR system (BioRad, USA), and each sample was tested in three parallels. β-actin served as an endogenous control. The results were analyzed with the 2^−ΔΔCt^ method.

### 2.8. Hematoxylin and Eosin (H&E) Staining and Immunohistochemistry

Placenta samples were embedded in paraffin after dehydration and sections were cut using a Leica RM2016 microtome (Leica, Shanghai, China), and then standard H&E staining and immunohistochemistry were performed. The detection antibodies used in immunohistochemistry were as follows: anti-CD31 (vascular marker) (Abcam, ab28364, Cambridge, MA, USA).

### 2.9. Statistical Analysis

Data in all figures and tables are expressed as mean with SEM. A two-way ANOVA was performed to compare more than two groups and the linear effects of H_2_S concentration and CD31 protein expression in the labyrinth zone area, placental area and placental weight using GraphPad Prism (GraphPad Software, La Jolla, CA, USA). *p* < 0.05 was considered to indicate statistical significance and different letters indicate significant differences.

## 3. Results

### 3.1. Diallyl Trisulfide Treatment Improved Litter Performance and Reduced Fat Deposition in Obese Mice during Pregnancy

We established a HFD-fed mouse model. The HFD treatment increased maternal body weight at 0.5 and 18.5 days of gestation, body fat percentage and inguinal fat content, while reducing maternal weight gain, the total feed intake and daily feed intake during gestation (Table 1). However, the DATS treatment reduced the inguinal fat content and body fat percentage of female mice. There was a significant interaction between diet and DATS treatment in the maternal inguinal fat content. In addition, the HFD treatment increased maternal litter size. The DATS treatment increased litter size and alive litter size but had no significant effect on the embryo absorption rate. These results indicate that the DATS treatment improved the litter performance and fat deposition of obese mice during pregnancy.

### 3.2. Diallyl Trisulfide Treatment Increased the Content of Hydrogen Sulfide in Obese Mice during Pregnancy

In order to investigate the effect of DATS treatment on the level of H_2_S, we analyzed the level of H_2_S in the maternal serum and placenta. We found that the HFD treatment reduced H_2_S levels in the serum and placenta (Figure 1A,B). However, DATS treatment increased H_2_S levels in the serum and placenta (Figure 1A,B). These results proposed that DATS treatment could improve the H_2_S level in the serum and placenta of mice.

### 3.3. Diallyl Trisulfide Treatment Improved Glucose Tolerance of Obese Mice

In order to investigate the effect of DATS treatment on the glucose tolerance of obese mice, we performed intraperitoneal glucose tolerance tests (IPGTTs). Compared with the NFD, the HFD treatment increased the blood glucose level and the area under the IPGTT curve in obese mice, while DATS treatment decreased the area under the IPGTT curve (Figure 2A,B). Furthermore, the concentration of blood glucose was significantly decreased in the HFD + DATS group after glucose injection for 60 min (Figure 2A,B). These results indicated that DATS treatment could significantly improve the blood metabolic parameters of obese pregnant mice and improve the glucose tolerance of obese mice induced by a high-fat diet.

### 3.4. Diallyl Trisulfide Treatment Promoted Placental Development

We analyzed the results of the fetus and placental development of mice in each group. We found that the HFD treatment reduced fetal weight (Figure 3A), while DATS treatment reduced placental weight (Figure 3B) and improved placental efficiency (Figure 3C). However, no significant interaction was observed between diet and treatment.

Next, we performed histomorphological analysis of the placentas by H&E staining and found that the HFD treatment reduced the placental area (Figure 3D,E) and the labyrinth zone area (LZ) (Figure 3F), but had no significant effect on the junction zone area (JZ) (Figure 3G). However, DATS treatment increased the placental area (Figure 3D,E) and the labyrinth zone area (Figure 3F), and there was a significant interaction between the placental areas. In order to determine the effect of placental H_2_S levels on fetal growth and placental development, we analyzed the correlation between placental H_2_S level and placental weight, placental area and labyrinth area. We found that H2S levels are positively correlated with the placental area (Figure 3I) and the labyrinth zone area (Figure 3J), but negatively correlated with the placenta weight (Figure 3H). These data indicate that DATS treatment promoted the placental development of obese female mice during pregnancy.

### 3.5. Diallyl Trisulfide Treatment Was Beneficial to Placental Angiogenesis

In order to investigate the effect of DATS treatment on placenta angiogenesis, we used immunohistochemistry and RT-PCR to detect the marker gene CD31 of placental vascular endothelial cells and the mRNA expression of angiogenesis-related factors. The results showed that an HFD inhibited the expression of CD31 protein (Figure 4A,B) in the labyrinth zone, while the expression of PLGF (Figure 4D), VEGFA (Figure 4E), VEGFR2 (Figure 4F) and HIF1α mRNA (Figure 4G) was upregulated. DATS treatment increased the expression of CD31 protein (Figure 4D) in the labyrinth zone of the placenta and upregulated the expression of CD31 (Figure 4C), PLGF (Figure 4D), VEGFA (Figure 4E), VEGFR2 (Figure 4F) and HIF1α mRNA (Figure 4G). In addition, the expression of CD31 protein in the labyrinth zone of the placenta and the mRNA expression of various angiogenesis-related factors have significant interactions between diets and treatments. In addition, through correlation analysis, we found that H_2_S levels are positively correlated with the expression of CD31 protein (Figure 4H) in the LZ. These data suggest that DATS treatment can promote placental angiogenesis of obese mice induced by a high-fat diet. The result is consistent with previous studies, which proposed that increased in placental efficiency may be attributed to better angiogenesis. Moreover, the pro-angiogenesis effect of H_2_S is also related to the upregulation of the HIF1α/VEGF pathway [31].

### 3.6. Diallyl Trisulfide Treatment Can Alleviate the Level of Placental Inflammation in Obese Mice during Pregnancy

In earlier studies, it was also found that back fat in the pregnant period of sows might cause toxicity of placental fat, which aggravates the inflammatory reaction of the placenta [32]. In order to investigate whether DATS treatment can alleviate placental inflammation, we analyzed the mRNA expression levels of *IL-10*, *IL-6* and *TNFα* in placental tissues and found that HFD could upregulate the relative expression of *IL-10* (Figure 5A), *IL-6* (Figure 5B) and *TNFα* mRNA (Figure 5C) in the placenta of mice. DATS treatment increased the relative expression of *IL-10* mRNA (Figure 5A) in the mouse placenta, but inhibited *IL-6* (Figure 5B) and *TNFα* mRNA (Figure 5C) expression. Moreover, *IL-10* and *IL-6* mRNA expression had a significant interaction between diet and treatment. These results suggest that DATS treatment could alleviate placental inflammation in obese mice during pregnancy.

### 3.7. Diallyl Trisulfide Treatment Improved the Lipid Metabolism in Placental Tissue of Obese Mice during Pregnancy

Next, we measured the lipid level and the expression of lipid transport related factors in the placenta. The results showed that the HFD treatment increased the levels of TG (Figure 6A) and LDL-C (Figure 6D). DATS treatment increased the levels of HDL-C (Figure 6C) but decreased the levels of LDL-C (Figure 6D). In terms of lipid transport, we detected mRNA expression of FABP4 and FATP4, and found that the HFD treatment increased the mRNA expression of FABP4 (Figure 6E) and FATP4 (Figure 6F), while DATS treatment increased mRNA expression of FABP4 (Figure 6E) and FATP4. In addition, TCHO and LDL-C levels, as well as mRNA expression of FABP4 and FATP4, had significant interactions between diet and treatment. These results suggest that DATS treatment can improve lipid metabolism in the placental tissue of obese mice during pregnancy.

## 4. Discussion

During pregnancy, the placenta is the only organ for connection between the mother and the fetus in mammals. The development of placental blood vessels is a key factor in determining the function of the placenta [33]. However, in both human and mouse models, it has been confirmed that maternal obesity during pregnancy can impair the development of the placental vascular system, leading to a decrease in placental blood vessel density [33,34]. Moreover, chronic inflammation caused by maternal obesity can lead to an imbalance of placental angiogenesis [21]. H_2_S, as an endogenous signaling gaseous molecule, is involved in the regulation of a variety of biological events in the animal body, such as promoting vascular development and relaxation, and anti-inflammatory and antioxidant effects [35]. Nevertheless, obesity affects H_2_S levels. Studies have shown that plasma H_2_S levels significantly decrease, and insulin resistance increases in obese or overweight individuals [36,37,38]. This is consistent with our findings.

Although the molecular link between placental function and H_2_S in obese individuals during pregnancy is unclear, it is clear that the upregulation of H_2_S levels plays an important role in cell-protective effects on insulin sensitivity, placental inflammation, lipid transport and placental angiogenesis. Our study found that HFD supplementation indeed reduced maternal insulin sensitivity and H_2_S levels in the serum and placental tissues, increased the concentration of placental inflammatory factors, and decreased placental vascular density in the LZ area. Moreover, a recent study also found that maternal obesity during pregnancy causes fetal growth restriction, molecular and structural changes in the transfer zone, which results in impaired trophoblast differentiation and placental dysfunction, despite increased maternal–fetal transfer capacity [39]. Maternal obesity can lead to systemic and placental inflammation. H_2_S, as a gas-signaling molecule, not only has an obvious anti-inflammatory effect at physiological concentrations, but also participates in the regulation of placental angiogenesis [40,41,42]. Our study found that DATS treatment increased maternal insulin sensitivity and H_2_S levels in the serum and placental tissue, upregulated the expression of anti-inflammatory cytokine IL-10 mRNA and inhibited the mRNA expression of pro-inflammatory cytokine IL-6 and TNFα in the placenta. In addition, the expression of placental vascular density and placental angiogenesis-related factors in the LZ area were upregulated. Consistent with this result, Bai et al. found that DATS could inhibit the activation of NF-κB and the expression of TNF-α in the colon tissue of patients with ulcerative colitis [43]. In the study of a renal ischemia–reperfusion (I/R) injury model, Su et al. found that NaHS alleviated kidney injury and dysfunction, apoptosis and the inflammatory response through Nrf2-mediated NLRP3 inflammasome inhibition [40]. In addition, in the early stage, our previous research found that H_2_S concentration of the maternal placenta serve as an important molecular mechanism that affects placental angiogenesis and piglet development [30], which is also consistent with our results. Therefore, we can conclude that DATS treatment can alleviate the inflammatory response in the placenta of pregnant obese mothers by regulating the expression of inflammatory factors, thereby improving placental angiogenesis.

Studies in humans have found that obesity during pregnancy increases the deposition of lipids into placental tissue, causing placental lipid toxicity [44,45,46]. Placental lipid toxicity can activate inflammatory signaling pathways and inhibit placental angiogenesis [47]. In this study, the HFD treatment increased the levels of TG and LDL-C, as well as the mRNA expression of FABP4 and FATP4, and decreased the level of HDL-C. DATS treatment decreased the levels of LDL-C, but increased the levels of HDL-C. Consistent with our findings, the exogenous H_2_S donor, NaHS, can improve the lipid metabolism of HFD-induced obese mice, reduce lipid deposition in the liver, and also can improve the body’s antioxidant capacity to reduce fatty liver [48]. Shi et al. found that allicin significantly increased the levels of TC and HDL-C, reduced the levels of LDL-C in the serum of obese mice, and reduced fat deposition in obese mice [17]. In addition, studies have found that increased lipid accumulation in the placenta leads to increased macrophage infiltration and decreased placental blood flow, leading to hypoxia and exacerbating placental inflammation, altering nutrient transport and leading to poor fetal health [18,19,20,22]. These data suggest that DATS treatment may improve abnormal placental fat transport by upregulating the expression of placental fatty acid transporters and regulate the level of placenta lipid metabolism. The reduction of placental lipid levels may be one of the reasons for alleviating placental inflammation and increasing placental blood vessel density.

## 5. Conclusions

H_2_S is an important signaling molecule in regulating angiogenesis. Gavage administration of DATS during the pregnancy of obese female mice increased alive litter size. In addition, the addition of DATS during pregnancy enhanced the lipid metabolism of the maternal, improved the insulin resistance of the obese mothers, and alleviated the placental vascular dysplasia caused by the obesity of the mothers.

## Figures and Tables

**Figure 1 nutrients-14-02230-f001:**
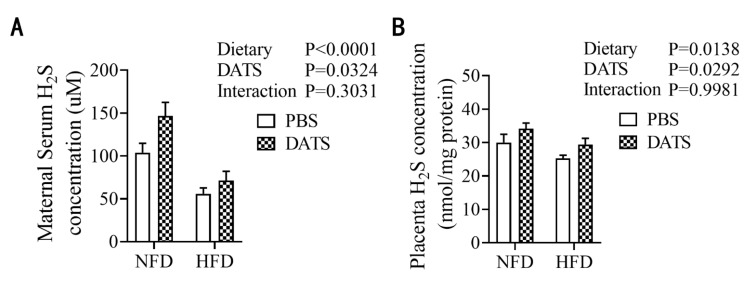
DATS treatment can increase the content of hydrogen sulfide in obese mice during pregnancy. (**A**) Serum H_2_S levels (*n* = 9–11 mice/group). (**B**) Placenta H_2_S levels (*n* = 7–10 mice/group). Two-way ANOVA—the values represent the mean ± SEM, *p* < 0.05 was considered to indicate statistical significance and different letters indicate significant differences.

**Figure 2 nutrients-14-02230-f002:**
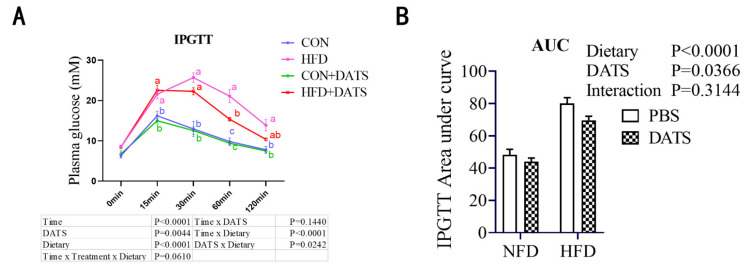
DATS treatment can improve glucose tolerance of obese mice. (**A**) IPGTT, 2 g/kg glucose (*n* = 6 mice/group). (**B**) The area under curve of intraperitoneal glucose tolerance tests (IPGTT). Two-way ANOVA—the values represent the mean ± SEM, *p* < 0.05 was considered to indicate statistical significance and different letters indicate significant differences.

**Figure 3 nutrients-14-02230-f003:**
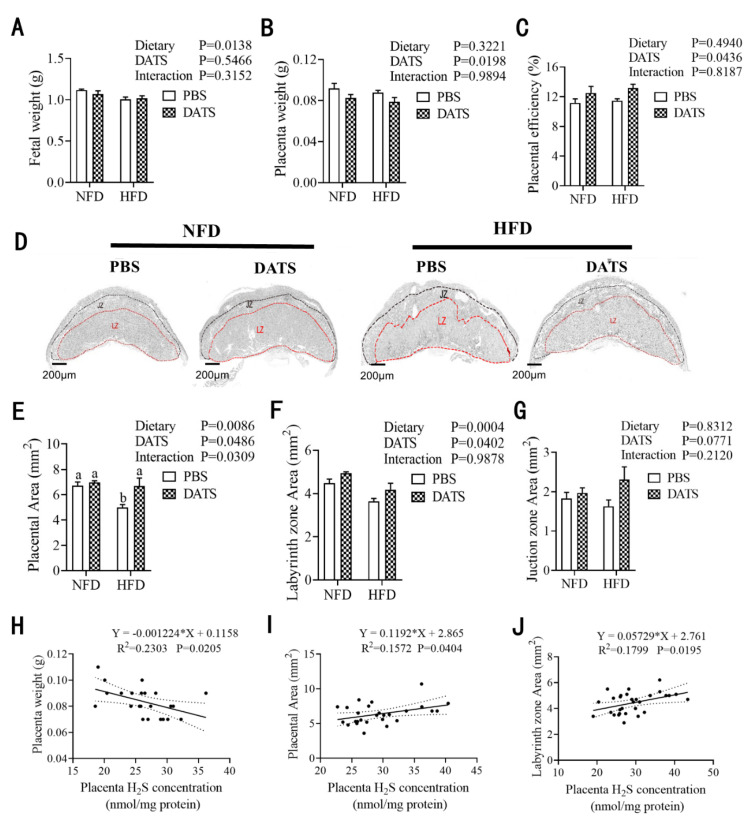
DATS treatment can improve placental development. (**A**) Fetal weight. (**B**) Placental weight. (**C**) Placental efficiency (fetal/placental weight ratio) (*n* = 9–11 mice/group). (**D**) The morphology of placental hematoxylin and eosin (H&E) staining (*n* = 8–11 placentas/group), scale bar 200 μm, the red dotted line region represents the labyrinth zone (LZ) and the black dotted line region represents the junction zone (JZ). (**E**) The area of placenta. (**F**) The area of LZ. (**G**) The area of JZ (*n* = 8–11 mice/group). (**H**–**J**) Correlation analysis of H_2_S levels with placental weight, placental area and LZ area (*n* = 8–11 mice/group). Two-way ANOVA—the values represent the mean ± SEM, *p* < 0.05 was considered to indicate statistical significance and different letters indicate significant differences.

**Figure 4 nutrients-14-02230-f004:**
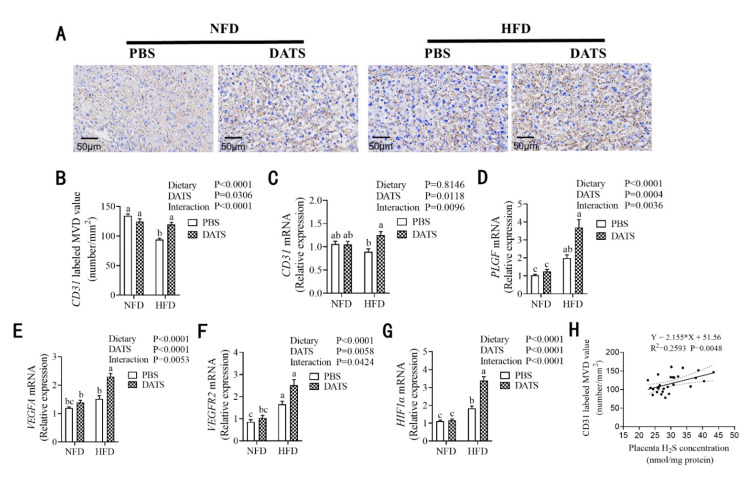
DATS treatment can improve placental angiogenesis. (**A**) Immunohistochemical analysis for placental CD31 staining in the labyrinth zone of the placenta (brown) (*n* = 8–11 placentas/group), scale bar 50 μm. (**B**) Expression of CD31 protein in the labyrinth zone (*n* = 8–11 mice/group). (**C**–**G**) Relative CD31, PLGF, VEGFA, VEGFR2 and HIF1α mRNA expression (*n* = 9–11 mice/group). (**H**) Correlation analysis of H_2_S levels with expression of CD31 protein in the labyrinth zone (*n* = 8–11 mice/group). Two-way ANOVA—the values represent the mean ± SEM, *p* < 0.05 was considered to indicate statistical significance and different letters indicate significant differences.

**Figure 5 nutrients-14-02230-f005:**
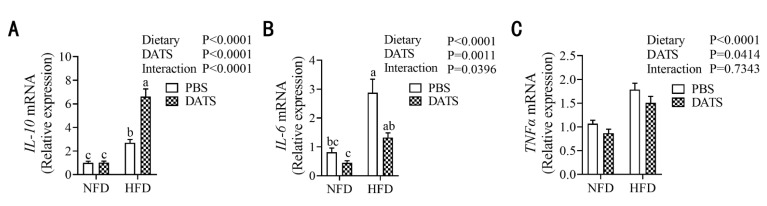
DATS treatment can alleviate the level of placental inflammation in obese mice during pregnancy. (**A**–**C**) Relative IL-10, IL-6 and TNFα mRNA expression (*n* = 9–11 mice/group). Two-way ANOVA—the values represent the mean ± SEM, *p* < 0.05 was considered to indicate statistical significance and different letters indicate significant differences.

**Figure 6 nutrients-14-02230-f006:**
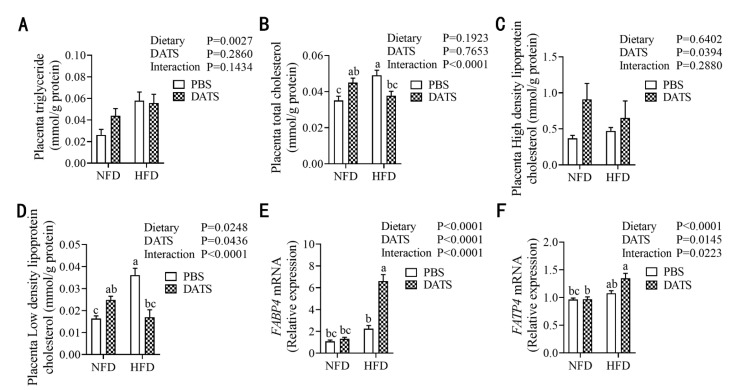
DATS treatment can improve the lipid metabolism of obese mice during pregnancy. (**A–D**) placental TG, TCHO, HDL-C and LDL-C levels (*n* = 9–11 mice/group). (**E**,**F**) Relative FABP4 and FATP4 mRNA expression (*n* = 9–11 mice/group). Two-way ANOVA—the values represent the mean ± SEM, *p* < 0.05 was considered to indicate statistical significance and different letters indicate significant differences.

**Table 1 nutrients-14-02230-t001:** Effects of DATS treatment on maternal characteristics and reproductive performance ^1^.

	NFD + PBS	NFD + DATS	HFD + PBS		HFD + DATS	*p*-Value		
						Dietary	DATS	Interaction
No.	9	11	11	10				
Maternal weights at E0.5 (g)	20.61 ± 0.23	21.22 ± 0.33	32.47 ± 0.46	33.09 ± 0.42		<0.01	0.12	1.00
Maternal weights at E18.5 (g)	34.07 ± 0.73	34.41 ± 0.68	40.40 ± 0.74	39.55 ± 0.69		<0.01	0.73	0.42
Gestational weight gain (g)	13.46 ± 0.56	13.18 ± 0.64	8.43 ± 0.37	6.47 ± 0.63		<0.01	0.09	0.07
Gestational total feed intake (g)	57.67 ± 0.60	56.79 ± 0.66	41.45 ± 1.87	39.73 ± 0.79		<0.01	0.27	0.97
Gestational daily feed intake (g)	3.21 ± 0.03	3.16 ± 0.04	2.31 ± 0.10	2.21 ± 0.04		<0.01	0.26	0.97
Gestational daily energy intake (kcal)	11.85 ± 0.12	11.67 ± 0.16	11.51 ± 0.52	11.04 ± 0.22		0.05	0.30	0.80
iWAT Percentage (%) ^2^	2.29 ± 0.09	1.76 ± 0.09	4.64 ± 0.15	3.98 ± 0.44		<0.01	0.01	0.77
Inguinal white adipose tissue weight (g)	0.77 ± 0.03 ^c^	0.62 ± 0.03 ^c^	1.92 ± 0.05 ^a^	1.44 ± 0.14 ^b^		<0.01	0.00	0.03
Litter size (No.)	7.78 ± 0.36	8.60 ± 0.43	8.64 ± 0.36	9.44 ± 0.24		0.02	0.03	0.98
Alive litter size (No.)	6.78 ± 0.55	8.40 ± 0.40	8.00 ± 0.47	8.50 ± 0.38		0.16	0.03	0.23
Embryo resorption rate (%) ^3^	6.50 ± 4.16	3.64 ± 2.44	7.47 ± 3.54	4.73 ± 2.23		0.76	0.40	0.99

^1^ Data were presented by mean value and standard error, and *p* < 0.05 was considered a significant difference. ^2^ The iWAT Percentage was calculated as: Inguinal white adipose tissue weight/body weight × 100. ^3^ The embryo resorption rate was calculated as: no resorption sites/no implantation sites × 100. ^a,b,c^ Different letters indicate statistical differences in the same row (*P* < 0.05)

## Data Availability

Data and the statistical code are available from the corresponding author upon reasonable request.

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
