# Peer review of "Diallyl Trisulfide Promotes Placental Angiogenesis by Regulating Lipid Metabolism and Alleviating Inflammatory Responses in Obese Pregnant Mice"

_nutrients, 2022, doi:10.3390/nu14112230_

Round 1

Reviewer 1 Report

The paper by Wang et al. is very interesting. The manuscript is well written.
Also, the statistical analysis is appropriately done.

Author Response

Thank you very much for your suggestion.

Reviewer 2 Report

Line 26: you stated that: “ In summary, our research revealed that H2S is an important signalling molecule that regulates angiogenesis.” In line 17 you claimed the same thus that is not quite new knowledge. Please in the abstract provide new information that you gained in the experiment.

Animal experiment: please clearly state how long the animals were treated with gavage. It was the cause for an additional suffer. I think that such manipulation should have been replaced by a simple addition to a diet of the bioactive or control preparation. What is the judgement for the applied dosage? Why did you use the regular chow? I am convinced that such fodder provide too much additional bioactive components. Of course you used the same chow in the control group but I think that semi-purified diet would be more appropriate.

Table 1: you presented that the interaction was shown only for “Inguinal white adipose tissue weight”, so only for that parameter the superscripts for the four groups should be shown. Why the superscripts in the following parameters were shown: Gestational total feed intake, Gestational daily feed intake?

Remaining tables and figures: when the interaction is >0.05 the all four groups should be not compared but only the main factors (two-way ANOVA rule).

Line 373: are available in [insert article here].???

Line 389: pregnancy = Pregnancy

Line 424, 429 etc: Capital letters use

Author Response

Thank you very much for your advice.The following is a point-to-point response to your comments.

Line 26: you stated that: “ In summary, our research revealed that H2S is an important signalling molecule that regulates angiogenesis.” In line 17 you claimed the same thus that is not quite new knowledge. Please in the abstract provide new information that you gained in the experiment.

Responses: Thanks! Revised, please see Line 27-28.

Animal experiment: please clearly state how long the animals were treated with gavage. It was the cause for an additional suffer. I think that such manipulation should have been replaced by a simple addition to a diet of the bioactive or control preparation. What is the judgement for the applied dosage? Why did you use the regular chow? I am convinced that such fodder provide too much additional bioactive components. Of course you used the same chow in the control group but I think that semi-purified diet would be more appropriate.

Responses: Thank you very much for your advice. The female mice was treated by gavage from E0.5 to E18.5. Please see Line 99-100. The doses of DATS was referenced to previous studies by Wu et al. (Phytomedicine 2011, 18:672-676).

If added through the diet is a very good way. However, our regular diet is a commercial diet and they cannot provide custom diet. If it's possible to follow up with similar studies we can make our own specific diets.

Table 1: you presented that the interaction was shown only for “Inguinal white adipose tissue weight”, so only for that parameter the superscripts for the four groups should be shown. Why the superscripts in the following parameters were shown: Gestational total feed intake, Gestational daily feed intake?

Responses: Thanks! Revised, please see Table 1.

Remaining tables and figures: when the interaction is >0.05 the all four groups should be not compared but only the main factors (two-way ANOVA rule).

Responses: Thanks! Checked and revised for tables and figures throughout the manuscripts

Line 373: are available in [insert article here].???

Responses: Thanks! Revised, please see Line 384

Line 389: pregnancy = Pregnancy

Responses: Thanks! Revised, please see Line 401

Line 424, 429 etc: Capital letters use

Responses: Thanks! Revised, please see Line 436,441

Reviewer 3 Report

The manuscript by Wang et al reported that Diallyl trisulfide (DATS) promotes placental angiogenesis by regulating lipid metabolism and inflammatory responses in obese pregnant mice. The study is well designed and has significant impact in the field, however, the manuscript should be undergone some revision before publication. I do have following comments and suggestion for authors.

Page 2, line 44-48: The authors should mention the study model (animal model/ human studies) where the finding was observed. Please specify what does “weak offspring” mean.

Page 3, line 122: Please mention the manufacturer of automated chemistry analyzer.

Page 4; Table 1: The authors mentioned a single SEM value for row. There should be SEM for each individual mean value.

Page 4, table 1: The authors should reconsider using superscript letters as same letter represent two different conditions in the same table. For example, c gives the information about the embryo resorption rate as well as statistical significance.

Page 4, table 1: Please reconsider using the term “body fat rate”.

Page 9, line 270-271: Authors mentioned that TNF-α mRNA expression had significant interaction between diet and treatment, however, in figure 5, P interaction is 0.7343.

Page 10, figure 6. In the axis titles, please spell check “liptein”.

Page 10, figure 6. Can you please clarify why the value of HDL-Cholesterol is higher than total cholesterol?

Page 10, figure 6: Can you please clarify why the cholesterol levels are normalized with protein instead of tissue weight?

Author Response

Thank you very much for your advice.The following is a point-to-point response to your comments.

Page 2, line 44-48: The authors should mention the study model (animal model/ human studies) where the finding was observed. Please specify what does “weak offspring” mean.

Responses: Thanks! Revised and added, please see Line 44-49

Page 3, line 122: Please mention the manufacturer of automated chemistry analyzer.

Responses: Thanks! Added, please see Line 123-124

Page 4; Table 1: The authors mentioned a single SEM value for row. There should be SEM for each individual mean value.

Responses: Thanks! Revised, please see Table 1.

Page 4, table 1: The authors should reconsider using superscript letters as same letter represent two different conditions in the same table. For example, c gives the information about the embryo resorption rate as well as statistical significance.

Responses: Thanks! Revised, please see Table 1.

Page 4, table 1: Please reconsider using the term “body fat rate”.

Responses: Thanks! Revised, please see Table 1.

Page 9, line 270-271: Authors mentioned that TNF-α mRNA expression had significant interaction between diet and treatment, however, in figure 5, P interaction is 0.7343.

Responses: Thanks! Checked and revised, please see Line 282

Page 10, figure 6. In the axis titles, please spell check “liptein”.

Responses: Thanks! Checked and revised, please see Figure 6

Page 10, figure 6. Can you please clarify why the value of HDL-Cholesterol is higher than total cholesterol?

Responses: Thank you very much for your reminder. HDL-C is all the lipids in the blood, while total cholesterol is the total amount of free cholesterol and cholesterol lipids in the blood. 

Page 10, figure 6: Can you please clarify why the cholesterol levels are normalized with protein instead of tissue weight?

Responses: Thank you very much for your comments. Placenta TG, LDL-C, HDL-C and T-CHO were detected using kits (Nanjing Jiancheng Bioengineering Institute, Nanjing, China). According to the instructions of the kit, animal tissues are generally adjusted with protein levels.